# In Vitro Salivary Protein Adsorption Profile on Titanium and Ceramic Surfaces and the Corresponding Putative Immunological Implications

**DOI:** 10.3390/ijms21093083

**Published:** 2020-04-27

**Authors:** Chen-Xuan Wei, Michael Francis Burrow, Michael George Botelho, Henry Lam, Wai Keung Leung

**Affiliations:** 1Faculty of Dentistry, The University of Hong Kong, Hong Kong, China; sanhin@connect.hku.hk (C.-X.W.); mfburr58@hku.hk (M.F.B.); botelho@hku.hk (M.G.B.); 2Department of Chemical and Biological Engineering, The Hong Kong University of Science and Technology, Hong Kong, China; kehlam@ust.hk

**Keywords:** ceramic, dental implants, immunology, salivary proteins and peptides, surface properties, titanium

## Abstract

Immune responses triggered by implant abutment surfaces contributed by surface-adsorbed proteins are critical in clinical implant integration. How material surface-adsorbed proteins relate to host immune responses remain unclear. This study aimed to profile and address the immunological roles of surface-adsorbed salivary proteins on conventional implant abutment materials. Standardized polished bocks (5 × 5 × 1 mm^3^) were prepared from titanium and feldspathic ceramic. Salivary acquired pellicle formed in vitro was examined by liquid chromatography-tandem mass spectrometry and gene ontology (GO) analysis to identify and characterize the adsorbed proteins. Out of 759 proteins identified from pooled saliva samples, 396 were found to be attached to the two materials tested—369 on titanium and 298 on ceramic, with 281 common to both. GO annotation of immune processes was undertaken to form a protein–protein interaction network, and 14 hub proteins (≥6 interaction partners) (coding genes: *B2M*, *C3*, *CLU*, *DEFA1*, *HSP90AA1*, *HSP90AB1*, *LTF*, *PIGR*, *PSMA2*, *RAC1*, *RAP1A*, *S100A8*, *S100A9*, and *SLP1*) were identified as the key proteins connecting multiple (6–9) immune processes. The results offered putative immunological prospects of implant abutment material surface-adsorbed salivary proteins, which could potentially underpin the dynamic nature of implant–mucosal/implant–microbial interactions.

## 1. Introduction

Implant-abutment interfaces are frequently associated with sustained peri-implant inflammation, characterized by significant inflammatory cell infiltrate and bone loss [1]. However, the exact microbiology, pathophysiology or the related host or parasitic immune reactions or interactions remain to be characterized [2,3]. The need for such an understanding has become more urgent, as implant restorations are continuing to be more widely used. According to the American Academy of Implant Dentistry [4], three million Americans had dental implants placed with an estimated growth of 500,000 per year. The market and cost for dental implants and implant-supported prostheses is anticipated to reach USD 9.1 billion by 2025 [5]. However, implants and their associated restorations are not resistant to biological complications such as dental-plaque-associated host responses that lead to the inflammation of peri-implant soft tissues [6,7], and the current longevity of dental implants has not surpassed that of successfully treated natural teeth [8]. Therefore, detailed understanding of bioactivity at implant–host tissue interfaces is still needed to improve the longevity of implants.

Recent efforts investigating interfaces of transmucosal dental implants or implant abutment materials claimed that biostability of peri-implant connective tissue and sulcular epithelium were critical to health maintenance [9], and the implant surface characteristics, such as surface topography and surface roughness, would modulate both osseointegration and the early formation of an effective connective tissue seal that would impair the downgrowth of epithelial cells [10,11]. Additionally, the human body is well-equipped to resist foreign materials/agents, and thus, dental implants and restorations might pose immunological challenges when placed in patients [2,3,12]. When dental implants or the associated restorations are placed in the oral cavity, an immediate protein adsorbed layer is formed from blood or saliva. This initial interface layer is considered to be a key determinant of how implant materials react with their environment at the molecular or cellular level [3]. In addition, the host immune response in relation to the initial adsorbed protein layer on a material might lead to subsequent favorable/unfavorable molecular, bacterial, or cellular interactions, and this, in turn, might influence the early healing of hard and soft tissues around implants and implant supra-structures [13]. The nature of selective protein adsorption to the implant surfaces, might lead to the concentration and composition of proteins on implant surfaces different from that found in free saliva, and as the very first host–material interaction after implant restoration placement, protein resorption is critical for the subsequent immunological behavior of implant materials [14]. For instance, genes representing immuno-inflammatory responses were found to be downregulated in cells adhered onto implants that were moderately roughened or had nanoscale topography 3–7 days post insertion [15]. The nature of the adsorbed proteins on implant surfaces causes specific functional alterations that might impact inflammatory processes [16]. As major reviews claim that the foreign body reaction might start from the initial protein adsorption to material surfaces, an understanding of how the adsorbed protein layers at supra- or sub-osseous compartments of dental implants transform the foreign body surface in the biological context is of significant interest [3,17].

On the other hand, bacterial colonization/biofilm formation at or close to the peri-implant sulcus triggers host immune responses, which might result in the development of peri-implant mucosal inflammation [7]. If lesions extend further apically, this might cause peri-implant bone loss, and thus peri-implantitis [8]. Hence, a better understanding of immunological functional aspects of the adsorbed protein complex at implant surfaces might be important in assisting interpretation of subsequent host reactions and biological behaviors of dental implant materials.

Titanium (Ti) and ceramic are conventional restorative materials for the reconstruction of a dental implant, and previous reports revealed a similar 5-year high survival rate (> 97%) for both ceramic and titanium implant abutments [18,19]. However, to date, information regarding interface events that occurred between these two biomaterials and salivary proteins remain scarce. Analysis of single protein adhesion to Ti and ceramic surfaces reported the presence of apolipoproteins, laminin V, glycosaminoglycans, collagenase, fibronectin, complement proteins, and fibrinogen [20,21]. However, the functional nature of the salivary protein complex interactions in terms of the host immune reaction and the comprehensive protein adsorption profile on Ti and ceramic is still not fully understood. Additionally, current interest in developing biocompatible materials with surface modifications for rapid healing and reducing microbial infection are also presumed to be influenced by the initially adsorbed surface proteins, yet the roles of these surface proteins in host responses and microbial attachment towards implant material surfaces remains unclear. As such, successful development of effective prevention or therapeutic approaches against bacterial infection of a dental implant and abutment, as well as improved integration/interaction with host soft tissues, might require a better understanding of the characteristics and biology of surface-adsorbed salivary proteins and in particular, their possible related immune functions or processes.

Putting the above together, exploration of how saliva or salivary components play a role in the immune system in particular, the early host response during implant wound healing, or at the prosthetic crown level, during the microbial biofilm formation, would help characterize the scientific basis underpinning the success of dental implants. The aims of this study were therefore, (i) to characterize the composition of adsorbed salivary proteins on Ti and ceramic surfaces, and (ii) to extrapolate the related immune and inflammation reactions of the adhered proteins that might influence early host response or microbial biofilm formation.

## 2. Results

### 2.1. Material Surfaces Characterization

Quantitative surface micro-roughness and surface morphology (Figure 1) of the Ti and ceramic surfaces after standardized polishing are shown. Standardized polished Ti and ceramic surfaces displayed similar average surface roughness and mean square roughness, with no significant differences detected (*p* > 0.05). R_ave_ and R_rms_ value of Ti surfaces were 0.139 ± 0.017 µm and 0.184 ± 0.025 µm, respectively, whereas for ceramic, it was 0.125 ± 0.027 µm and 0.168 ± 0.039 µm. Qualitatively, the Atomic-force microscope (AFM) scan images and corresponding surface profile scans highlighted no major three-dimensional topographic differences between the two materials. 

### 2.2. Surface Energy

Surface energy, calculated from the contact angles (Figure 2) based on the Good–van Oss theory (vOGT), was assessed by total surface free energy and values of surface free energy components (mJ/m^2^) of all specimen groups. The results exhibited relatively similar surface free energies between the tested material surfaces, without any statistical difference (Table 1).

### 2.3. Adsorbed Proteins Characterization

Bicinchoninic acid (BCA) assay was applied to assess the protein concentration of filter-sterilized unstimulated whole saliva samples, before the material specimens surface film formation. The same protocol was applied for measurement of the total adsorbed salivary protein quantity on each specimen. Protein content of the saliva samples from healthy donors was 0.82 ± 0.09 mg/mL, adsorbed protein on Ti and ceramic were 1.09 ± 0.45 μg or 1.90 ± 0.40 μg/specimen, respectively (*p* > 0.05). Overall, 386 proteins were identified (Appendix A), with 281 proteins in common between the two materials. There were 88 proteins detected exclusively on Ti and 17 on ceramic.

PANTHER protein classification (Table 2) revealed that among the detected proteins, compared to filtered unstimulated whole saliva, nucleic acid binding proteins and enzyme modulator proteins occupied relatively higher proportions (>10%). Apparently, defense/immunity protein (PC00090) on both materials (Ti/ceramic: 35/33 of 38) or cell adhesion molecules (PC00069) on Ti alone (14 of 17) appeared enriched while less or no transferase (PC00220) or membrane traffic protein (PC00150) on both materials (Ti/ceramic: 9/7 of 34 or 0/0 of 10) were detected, respectively (Table 2). Viral protein (PC00237) appeared readily attachable onto both tested materials (Table 2).

### 2.4. Ontological Network Reconstruction and Analysis of Adsorbed Proteins

After examining the related biological processes, a further query of the whole identified protein ID lists for Ti and ceramic groups were put into databases in the ClueGO + CluePedia plugin to construct a comprehensive protein–protein interaction network regarding immune system processes, with 159 related proteins and 14 functional terms (Figure 3).

In order to identify and characterize representative genes that function as key connections between functional terms, the connectivity degree distribution (connectivity between nodes) of the constructed protein–protein interaction network (Figure 3) were examined, and proteins with six or more direct interacting partners were selected as essential proteins in the network, based on the previously proposed methods [24,25,26]. Fourteen hub proteins were identified (coding gene: B2M, C3, CLU, DEFA1, HSP90AA1, HSP90AB1, LTF, PIGR, PSMA2, RAC1, RAP1A, S100A8, S100A9, and SLPI), which represented 30.5% of the total protein connectivity. The constructed sub-network revealed the possible relations of these proteins and related immunological processes (Figure 4).

## 3. Discussion

### 3.1. Surface Adsorbed Proteins

Ti and ceramic, when exposed to the oral environment, induced sequences of interactions with the oral biological components, starting with protein adsorption [27,28]. In the oral cavity, the initial adsorbed salivary proteins formed an interface between the restoration surface and the subsequent attached proteins, cells, or microorganisms. Salivary proteins on the transmucosal part of titanium dental implants claimed to be involved in epithelial adhesion and inhibition of oral bacteria [29]. Previous studies on the influence of salivary protein films on the quantity of initial *Candida albicans* adhesion on titanium and ceramic implant surfaces found that sand-blasted titanium (relatively higher in surface roughness) produced the lowest luminescence intensities for both salivary mucin and albumin, as compared to machined and sand-blasted titanium; while acid-etched titanium, zirconia ceramic, and saliva mucin were found to promote *C. albicans* adhesion, and albumin was found to act as a blocking agent [30]. Additionally, the presence of saliva pellicles was found to greatly reduce the number of *Streptococcus mutans* bound to titanium and zirconia ceramic surfaces [31]. Interestingly, the present study observed that salivary defense/immunity protein appeared to more readily adhere onto Ti or ceramic surfaces (Table 2), which might possibly contribute to explaining this phenomenon. The nature of such enrichment or ‘specific adhesion’, however, remains to be elucidated.

The material surface energy and topographic properties play an important role in the formation of the initial protein layer [32]. Teughels and co-workers (2006) reported that an increase in surface roughness or Ra above the threshold of 0.2 μm, facilitates biofilm formation on restorative materials, however, the existing evidence raised an inconclusive relationship between surface roughness and protein adsorption [33]. In the present study, focus was given to the protein adsorption profile and related immunological function, and thus, an attempt was made (Ra < 0.2 μm) to minimize the influence of the topographical properties of Ti and ceramic with standardized polishing. Considering the complicated nature of body-fluid–biomaterial surface interactions [34], the current study, in no way was able to decipher the mechanism underpinning salivary protein adsorption or interaction on Ti or ceramic surfaces. This report rather, at best, served as a sound step forward towards the beginnings of understanding such interactions.

Additionally, GO analysis was adopted in this study to examine the potential implant integration related to immune processes contributed by the surface-adsorbed proteins. Such a method has been previously used by Thalji and coworkers [15] in comparing implant surfaces adherent cell gene expression profiles that were differentially regulated by various implant surfaces, which to our understanding, is the first group to carry out GO analysis of osseous level implant attached cell RNA data.

From the protein quantity measurement, only around 0.1%–0.2% of proteins were adsorbed from the saliva sample added to each material. From the protein identification test, and out of the 759 proteins identified from the pooled saliva samples, less than half were found to be attached to either of the material surfaces, indicating putatively a selective adsorption process from liquid to solid surfaces, as documented previously [33]. At the time when the mass spectrometry and protein identification experiments were carried out, the protein quantification function of the MaxQuant software was not available to the study team, which unfortunately prohibited any corresponding quantitative analysis. Nevertheless, the qualitative data showed proteins detected in the saliva samples used in this study were mostly nucleic acid binding proteins, enzyme modulator proteins, hydrolase, cytoskeletal protein, oxidoreductase, signaling molecule, calcium-binding protein, and defense/immunity protein, based on their classification in descending order (Table 2). As highlighted in previous sections, results of this preliminary in vitro investigation appear to suggest that selective salivary protein adsorption on Ti and ceramic surfaces might be possible. Further investigations, however, are needed to confirm the above speculation.

Previous efforts in identifying surface-adsorbed proteins on Ti and ceramic mostly targeted individual protein groups, such as glycoproteins, haptoglobin, hemopexin, apolipoproteins, vitronectin, fibronectin, and fibrinogen [20,35]. In this study, we were able to provide a more detailed profile of adsorbed salivary proteins, with high throughput proteomic techniques. The surface-adsorbed proteins detected, reflected a variation in adsorption affinity between different material substrata at the 2-h in vitro adsorption point (Appendix A). One possible contribution to the variation might be the different physiochemical nature of the substrata that affected the initial protein adsorption affinity and the subsequent competitive exchange and aggregation, known as the Vroman effect [36]. The current preliminary report, described in a static in vitro setup/model, the possible interactions or interrelationships between Ti/ceramic and filtered unstimulated whole saliva, in particular the proteins related to immune responses (are to be discussed in Section 3.2.). Knowledge about the in vitro adsorbed salivary proteins could inform the design of future experiments to explore the physicochemical properties/interactions of acquired salivary pellicle on oral non-shedding surfaces, healthy or diseased. The dynamic nature of *in vivo* oral/dental biomaterial surface–saliva interactions, in line with the concepts underpinned by Vogler [34] for cardiovascular biomaterials and blood-proteins, remained to be explored. 

### 3.2. Immune Responses Presented in the Network

Evidence from gene expression studies in humans indicate that genes representing an immuno-inflammatory response are overexpressed during the initial stage of implant healing, followed by genes representing osteogenic processes, bone remodeling, angiogenesis, and neurogenesis [15,37]. Hence, host immune reaction plays a vital role in the subsequent implant osteointegration and overall implant restoration biocompatibility [38]. Previous investigations claimed that adsorbed proteins on the biomaterial surface were able to provoke host immune responses and regulations, yet little has been reported on the process through which this protein layer influences the related immune processes [39,40]. In an attempt to obtain additional information in this context, integrative ontological networks regarding immune system processes were built with all identified surface proteins. 

Even though the materials used were clinically non-immunogenic, nontoxic, and chemically inert, as a foreign body they might still trigger inflammatory responses through, for instance, nonspecific binding of antibodies to biomaterial surfaces that might trigger the activation of the immune complement system [41], and subsequently activate pro-inflammatory mediators that pertain to acute inflammation [40,42]. The related processes were also manifested in the network models of the present study, as the material surface-adsorbed immunoglobulin components were able to trigger and regulate complement activation (Figure 3 and Figure 4). One example of biomaterial induced inflammation is the foreign body reaction, which is the primary reaction of the nonspecific immune system evoked through contact with foreign materials [38]. Proteins such as fibrinogen, complement, and antibodies found on both material surfaces in the present study were previously claimed to be involved in the onset of foreign body reaction, through processes similar to the natural inflammation reaction in wound healing but were altered by the nature of the foreign material [43]. The constructed network in this study revealed putative roles of fibrinogen, complement component 3 (C3), and immunoglobulin, in the onset and progression of the host immune responses that might share the same pathway of foreign body reactions. 

Moreover, it has also been documented that surface adsorbed proteins could promote or inhibit bacterial attachment [40]. The existing evidence proved that some specific salivary proteins present on the Ti or ceramic surfaces in this study like collagen, fibrinogen, fibronectin, and elastin, function as receptors for bacterial surface adhesins [44,45]. Despite the individual protein–microbe interactions, the constructed network in this study revealed the possible collective functional linkages between these surface-adsorbed proteins in host immune processes. For instance, an antibody-independent process activated by microbes (classical pathway) could lead to the activation of the complement system for recognizing and eliminating invading microorganisms that were manifested in the network (Figure 3) as links between antibacterial/antimicrobial humoral responses, which could further complement activation by binding antibodies complexed with antigens [46]. 

### 3.3. Hub Proteins

Previously, Thalji and coworkers [15] adopted GO analysis comparing implant surface adherent cell gene expression profiles on various implant surfaces of related biological classes, however, the differentially upregulated genes found were mostly related to cellular/extracellular components. In the present study, filtered saliva was adopted to minimize the effects of oral microbes and cells in saliva, and thus, the GO analysis was able to narrow down to the category of immune system process, to explore the protentional host-material immunological interaction contributed by the surface-adsorbed proteins. Subsequently, in order to further identify the key proteins within the above constructed protein network, hub protein analysis was applied. 

To date, no clear standards regarding hub proteins determination has been established, but it is clear that highly connected proteins are more likely to be essential for the stability of the network compared to their non-hub counterparts, as they might serve as central communication points across a network due to their high connectivity [24]. In this study, some common proteins shared between Ti and ceramic were observed to link several immune processes in the same biological network. For instance, the proteins DEFA1 (defensin, alpha 1) and LTF (lactotransferrin) linked the antimicrobial and antibacterial humoral responses to the activation of leukocytes and myeloid cells, and toll-like receptor (TLR) signaling pathway, the combination of which might trigger complement activation and possible advanced immune responses. After key protein identification and sub-network construction (Figure 4), Ti and ceramic were found to share most hub proteins and all related immune processes, which implies the possibility that the saliva conditioning proteins on both materials might share some similar surface immune defense mechanisms. Additionally, most of the hub proteins presented in this study were individually identified previously as important inflammatory modulators, for instance, C3 and S100 calcium-binding protein A9 (S100A9) that could modulate host inflammatory reactions or regulate myeloid cell function by binding to TLRs, resulting in the onset of inflammatory responses to terminate infectious factors [39,47]. In this study, to the best of our knowledge, the salivary immune components adhered on Ti and ceramic were for the first time linked together through interaction networks to reveal the possible underlying immunological activities on Ti and ceramic surfaces. 

The network constructed with 14 hub proteins also demonstrated the inter-linked role of the corresponding cascade reactions of antimicrobial humoral responses, leukocyte, and myeloid cell activation and related immune responses, complement activation, and Fc receptor signaling pathways. As the primary host immune reactions toward implant material surfaces are governed by nonspecific defense systems, the activation and response of the complement system plays a pivotal role in host immune response in foreign body reactions of biomaterials, which might lead to monocyte/macrophage activation and migration [13,39]. However, no general agreement has been made regarding whether or not implant restoration surfaces would benefit from this complement activation [41]. Alternatively, the constructed network might also function in a manner of immune response activation in response to microbial products and immune complexes. For instance, the presence of hub proteins S100A9, DEFA1, and LTF could serve as the link between antimicrobial humoral response, TLR signaling pathway, and myeloid and leukocyte activation. As the immune response progresses, the subsequent activated complement or antibody could coat a pathogen and then be removed by the migrated phagocytes or cytotoxic cells [46]. As such, the network constructed revealed possible host immune responses against both foreign bodies as the material itself, and colonized microbes on the Ti and ceramic restoration sites. The knowledge gained could further contribute to the understanding of the underlying pathogenesis of Ti and ceramic restoration related to clinical complications.

### 3.4. Saliva Proteins in Health and Diseases

Saliva flow rate and ionic composition is affected by the autonomic nervous system while the organic contents such as enzymes, various host defense molecules, growth factors, and inflammatory mediators appear to be regulated by hormones of serval endocrine systems [48,49]. Alterations in the host’s health state might alter the proteomic composition of saliva [50]. Periodontal disease, for instance, is associated with lower antioxidant capacity in whole saliva and increased protein oxidation [51]. To date, many salivary proteins have been found to be elevated in periodontal diseases and have been identified as biomarkers, such as interleukin-1β and matrix metalloproteinase-8 [52]. Previous study comparing the saliva protein components of healthy and periodontal disease individuals found increased serum albumin, hemoglobin, immunoglobulin, and lower abundance of cystatin in the saliva of those with periodontal disease [53]. Higher salivary IL-6 and IL-10 levels were found among edentulous patients with implant biological complications, perhaps indicating the association between salivary cytokines with implant-related disease [54]. It remains to be seen if such alterations in salivary cytokine content associate with alteration of salivary protein and then directly or indirectly correspond to the implant surface-adsorbed protein compositions.

Total saliva protein concentration was also found to be higher in caries-susceptible individuals than caries-resistant individuals [55]. Previous work that recruited caries-free and a caries-susceptible subjects to study the correlation between saliva peptide composition with dental caries susceptibility, suggested higher proteolytic activity and lower amounts of phosphopeptides, histatins, and statherin, would increase the susceptibility to dental caries [56]. However, to date, no report concerning the nature and content of salivary proteins has been reflective of caries-prone individuals, however, the relevancy of saliva proteins and corresponding salivary pellicle defense remains to be elucidated.

Additionally, saliva flow rate also affects its proteomic composition, the protein composition of saliva at high flow rates more closely resembles that of the primary saliva produced by the acinar cells, whereas saliva at low, unstimulated flow rates tends to have more protein variations [57]. Hence, when investigating unstimulated whole saliva, such variations should be noted in patients with hyposalivation, or xerostomia. For instance, dental implants in oral cancer patients who went through irradiation were previously observed with peri-implant bone loss, which might suggest the negative immune/defense associated with hyposalivation [58].

As predicted, immune processes and pathways in the form of protein–protein interaction networks on Ti and ceramic surfaces were static structures based on both experimental evidence and the computational assumption of the co-regulation genes [59]. Hence, further intraoral experimental confirmation of the presence and interaction fashion of surface attached salivary proteins on Ti and ceramics are still needed. Additionally, possible variations should be included in future investigations, such as variations in saliva of healthy or diseased individuals (such as individuals with caries or periodontitis), patients with low saliva flow rate (such as dry mouth or impaired salivary function), and implant health.

### 3.5. Limitations

A closed, static-filtered, unstimulated, whole saliva–biomaterial interaction model was employed in this study to minimize the potential influences from oral microbes and their proteins. In addition to the fact that Ti and ceramic are known to be biologically inert materials, the in vitro interactions observed most likely were not biologically driven. The results reported in this study therefore need to be interpreted cautiously.

The act of filtering could remove large salivary molecules and hence could reduce the total protein amount in the saliva sample, and to some extent, the quality of the protein to be attached onto the material surfaces tested, therefore, caution in interpretation of the current data is recommended. Nevertheless, to our best knowledge, we identified more surface-adsorbed proteins on titanium or ceramic than previously reported. The next limitation was the nature of the complexity of the oral environment, the dynamic protein adsorption/dehydration process [34] and the relatively low protein quantities available for extraction and characterization, which greatly limited the investigations of the material surface-adsorbed protein complex. Mass-spectrometry-based proteomics, therefore, become the method of choice for complex protein analyses, however, due to technical and inter-individual variations, comparison of pellicle proteome datasets between laboratories often remains sub-concordant [60]. The current available information from high throughput proteomic experiments enabled the construction of large-scale protein networks for automatic protein function prediction and annotation, yet the lack of standardized protocols for data processing, data base searching, and data sharing among different databases, platforms, and software [61] created challenges in the pellicle proteins study, especially for biologists and clinical investigators. *In vivo* investigations designed to quantify acquired proteins identified, as well as designed to explore specific, dynamic, saliva–dental/biomaterial surface adsorption and interactions are warranted.

## 4. Materials and Methods

### 4.1. Specimens Preparation

Block-shaped specimens (5 × 5 × 1 mm, 70 mm^2^) were made from pure grade 2 Ti (Permascand, Ljungaverk, Sweden) and a feldspathic ceramic for computer-aided design and computer-aided manufacturing (CAD/CAM) restoration fabrication, shade A3.5 (Vitablocks Mark II™, VITA, Bad Sakingen, Germany). The surfaces of all specimens were wet-polished at room temperature (Ecomet 5, Buehler, Lake Bluff, IL, USA) at 100 revolutions/min (rpm), using 1200-grit silicon carbide abrasive papers, for 3 minutes. Specimens were cleaned in an ultrasonic bath with distilled deionized water (pH 7.0) for 10 min, to remove any debris from the polishing procedures. All specimens were immersed in 70% ethanol for a period of 3 days for disinfection, and rinsed with and stored in sterile distilled water for 12 h before use.

### 4.2. Material Surfaces Characterization 

Five specimens prepared from either Ti or ceramic were randomly selected for surface morphology characterization using an AFM (NanoScope 8 MultiMode AFM, Bruker Nano Inc., Nano Surfaces Division, Santa Barbara, CA, USA). The measurements were performed in contact mode using silicon nitride levers (Bruker SCANASYST-AIR, USA) with a measurement area of 100 × 100 µm^2^, and the scan format was set at 256 × 256 pixels. The central positioning area of each specimen was selected for analysis, and care was taken to relocate to the same area during measurements for both material groups. Digital images of the original measurements were used for morphological description with Bruker NanoScope Analysis software (Version 1.40). Two parameters, average roughness (R_ave_) and root mean square roughness (R_rms_) were adopted to characterize the surface topography. 

The estimated surface free energy of the materials was analyzed with a Contact Angle Meter (DM-701, Kyowa Interface Science Co., Ltd., Tokyo, Japan) using three probe liquids (0.5–1 μL/drop)—de-ionized water, diiodomethane, and ethylene glycol at 21 °C, and relative humidity of 21%. The average contact angle values were then analyzed for surface free energy (mJ/m^2^), using vOGT [62].

### 4.3. Acquired Salivary Protein Films Formation 

Saliva collection protocol was approved by the Institutional Review Board of the University of Hong Kong/Hospital Authority Hong Kong West Cluster (UW 17-075) and written informed consent obtained from the donors. Unstimulated whole saliva was collected directly prior to the experiments by expectoration from five healthy donors aged 24–33 years (three females and two males), who were nonsmokers with no systemic disease, were not taking any medication, and were free from gingivitis (full mouth bleeding on probing score < 20%, and caries (decayed, missing, and filled permanent teeth = 0). Before collection, donors refrained from eating and oral hygiene for at least 2 h. The saliva samples were collected between 9 and 10 AM using an ice-chilled container, and were immediately used after sterilizing through single-use filtration devices (0.22 µm, Steritop, Millipore Corporation, Billerica, MA, USA), to remove debris and microorganism from the saliva.

The Ti or ceramic specimens were put into a 48-well plate (IWAKI, Tokyo, Japan), and in situ pellicle formation took place by incubating each sample block with 1 ml of filter-sterilized saliva at 37 °C, for 2 h, with gentle agitation. The specimens were then removed and each specimen was dip-washed twice in 1 ml ultra-pure milliQ-H_2_O, to remove loosely bound components.

The current study did not attempt to investigate the mecahnism driving protein adsorption or saliva protein–Ti/ceramic surface interaction.

### 4.4. Quantitative Analysis of Adsorbed Surface Proteins 

Ten saliva-coated specimens were taken from each material group, and 200 µL of radio-immunoprecipitation assay buffer (RIPA, Thermo Fisher, Rockford, IL, USA) was added to each group for lysis of the attached protein. The dissolved protein concentrations, as well as the protein concentration of the saliva sample before incubating with material blocks were determined using the Thermo Scientific™ Pierce™ BCA Protein Assay Kit (Thermo Fisher Scientific, Rockford, IL, USA), as described previously [63,64,65]. The protein concentrations (µg/material specimen) were expressed as means ± SD. Bovine serum albumin dissolved in the RIPA buffer was serially diluted to similar concentration ranges and used as a standard. 

### 4.5. Mass Spectrometry and Proteins Identification

Following standard shotgun proteomics protocols, approximately 15 μg of each protein sample was dissolved in 50 µL of 8 M urea in 100 mM Tris-HCl (pH 8.5), for denaturation, at 60 °C for 10 min, followed by disulfide bond reduction with dithiothreitol (20 mM), for 20 min at room temperature, and alkylation with iodoacetamide (25 mM) at room temperature, for 30 min, in the dark. The resulting protein solution was diluted with 100 mM Tris (pH 8.5) to reduce urea concentration to 1 M, followed by in-solution trypsin digestion at 37 °C for 16 hours, with a trypsin/protein ratio (*w*/*w*) of 1:50. After acidification, the digests were collected, centrifuged at 14,000 rpm for 30 min, desalted with ZipTip C18, and speed vacuum dried. The digests were re-dissolved in 0.1% formic acid, prior to liquid chromatography tandem–mass spectrometry (LC–MS/MS).

Nanoflow high-performance liquid chromatography coupled with a Linear Trap Quadropole Orbitrap Velos mass spectrometer (Thermo Fisher Scientific, Waltham, MA, USA) was used for analyses. In brief, all samples were loaded onto a reverse-phase chromatograph, a PicoTip column (New Objective, Woburn, MA, USA) (360 μm outer diameter, 75 μm inner diameter, 15-μm tip), which was packed with 8 to 10 cm of octadecyl-silica-A C_18_ 5-μm beads (YMC America Inc., Allentown, PA, USA). The peptides were rinsed for 5 min with solvent A (0.1% formic acid) and eluted into the mass spectrometer with a 150-min linear gradient from 2% to 35% of solvent B (100% acetonitrile in 0.1% formic acid). The fragmentations of peptide ions were obtained in a data-dependent mode—each cycle consisting of a full scan followed by 20 collision-induced dissociation MS/MS scans on the 20 most abundant ions from the immediate preceding full scan. Data were acquired with two replications of the same sample, to ensure more complete coverage.

The MS data were processed using the MaxQuant software (version 1.5.2.8; Max Planck Institute of Biochemistry, Martinsried, Germany) [66]. Sequence database search was performed by the Andromeda search engine in MaxQuant against the *Homo sapiens* reference database (UniProt). Default parameters were adapted for protein identification. Trypsin/P was specified as the digestion enzyme. The mass tolerances of the precursor and fragment were set at 20 ppm and 0.5 Da, respectively. Variable modifications were set to include acetylation (on protein N-terminus) and oxidation (on methionine residues), and a fixed modification of carbamidomethylation (on cysteine residues). The minimum peptide length was set to seven amino acids, and a maximum of two missed cleavages was allowed. A maximum false discovery rate of 0.01, estimated internally by MaxQuant, was applied for both peptide and protein identifications. Quantification of the identified proteins was not available during the data acquisition.

### 4.6. Data Mining from Identified Proteins

The extracted surface conditioning proteins involved in immune system processes were identified. The acquired protein gene lists of the four groups were functionally annotated with the Cytoscape (v3.5.1) [67] plugin Cluego (v2.5.4) + CluePedia (v1.5.4) [68], by mining their associations with previously-defined GO terms, downstream from the immune system processes (GO:0002376) in which they might be involved or localized. The resultant GO terms were decoded and visualized in the groups of the immune system process network. The default parameters were adopted for GO term restrictions and set at 4% of genes with a *p*-value < 0.05, which meant at least 4% of genes in the gene cluster were significantly correlated to the GO term. GO term connectivity (kappa score) was set to 0.4, as the threshold. Subsequently, the hub proteins were determined with Cytoscape inbuilt network analysis of connective degree distribution. The cutoff was selected based on the distribution of first-degree neighbors to identify hub proteins, as outlined previously [24,25,26]. The corresponding sub-network was constructed by isolating the non-hub neighbors, to examine interactions between the hub nodes and related immunological functions.

### 4.7. Data Analysis

Data collected were analyzed using the statistical software IBM SPSS Statistics 20 (SPSS, Armonk, NY, USA). *p* < 0.05 was considered to be statistically significant. 

## 5. Conclusions

Within limitations of this study, the present in vitro investigation reported qualitatively pooled unstimulated salivary protein adsorption profile on titanium and ceramic surfaces, characterized via ontological networks immune system process functions/responses. Whilst the in vivo mechanism and physicochemical properties of such phenomena remain to be elucidated, the possibility of the surface adsorbed salivary proteins involvement in immune processes at the biomaterial soft-tissue interface might be considered an important subject relevant to implant dentistry; in particular, the host immune processes tested against the implant biomaterials and the former tested against oral microorganisms. This offers new insights towards understanding the pathogenesis of Ti and ceramic restoration associated clinical reactions or complications. The dynamic nature of *in vivo* Ti or ceramic surface-adsorbed saliva proteins interactions requires further exploration.

## Figures and Tables

**Figure 1 ijms-21-03083-f001:**
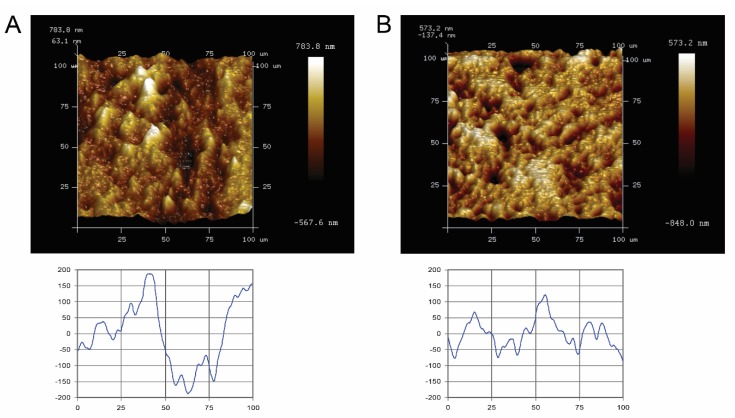
Atomic-force microscopy (AFM) images of representative specimen surfaces (100 × 100 µm^2^, upper panel), and corresponding scan profiles (lower panel) of the tested blocks. (**A**) titanium and (**B**) ceramic. Bright areas indicate high points, and dark areas indicate low points across the surface. (Lower graph: X- and Y-axes measured in micrometers).

**Figure 2 ijms-21-03083-f002:**
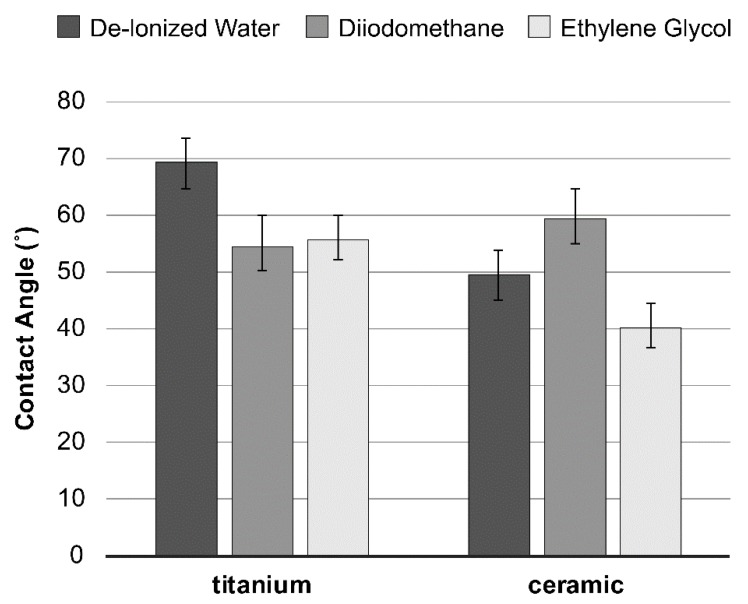
Average contact angles (± SD) of the three probe liquids on the material tested.

**Figure 3 ijms-21-03083-f003:**
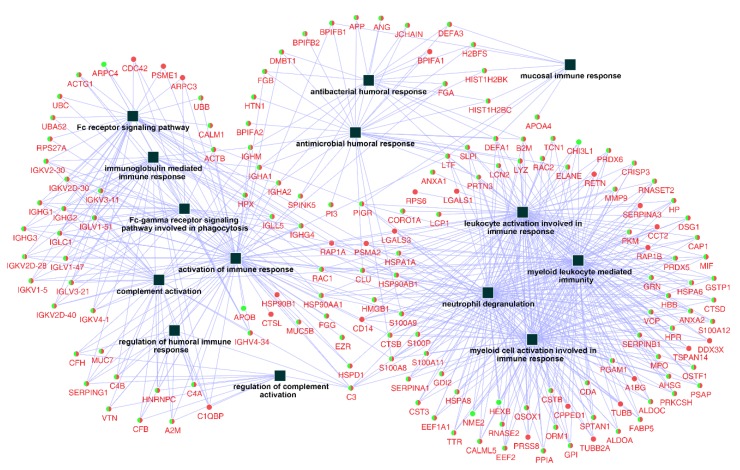
Network of the immune system process constructed by identified proteins on titanium and ceramic surfaces. The list of 386 identified proteins were imported into Cytoscape to reconstruct the immune system process network, resulting in 159 related protein nodes and 14 functional terms (squares). Round nodes with green (Titanium) and red (ceramic) colors were the common proteins between the two groups, and the functional terms are shown with square nodes.

**Figure 4 ijms-21-03083-f004:**
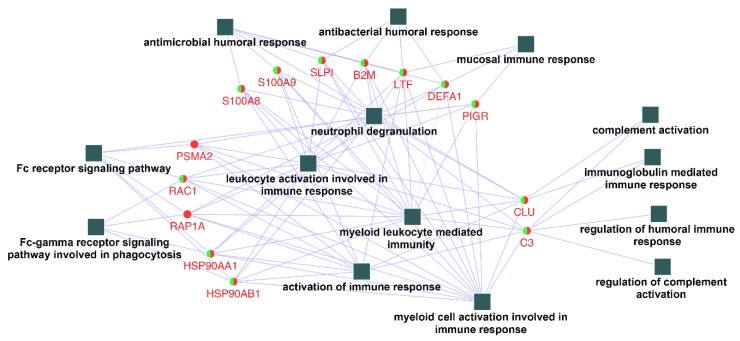
Networks contracted by the 14 hub proteins selected and their related functional terms identified (squares, c.f. Figure 3). After connection degree analysis, a list of 14 proteins (degree ≥ 6) were selected and imported to Cytoscape to reconstruct the ontological network. All fourteen biological process terms (square) were found to be directly connected to the hub genes (circles).

**Table 1 ijms-21-03083-t001:** Surface free energy (mJ/m^2^) of the tested specimens determined by the Good–van Oss theory (vOGT).

Specimen	Total Surface Free Energy	Non-Polar Component	Acid-Base Component	Acidic Contribution	Basic Contribution
Titanium	31.6	31.6	0.0	19.3	0.0
ceramic	37.8	29.0	8.8	38.8	0.5

**Table 2 ijms-21-03083-t002:** Classification of detected proteins.

Category Name ^a^	Adsorbed on Titanium #hits	Adsorbed on Ceramic #hits	Filtered Unstimulated Saliva #hits
nucleic acid binding (PC00171)	90	65	207
enzyme modulator (PC00095)	62	50	96
signaling molecule (PC00207)	40	36	57
cytoskeletal protein (PC00085)	36	24	68
hydrolase (PC00121)	34	33	84
defense/immunity protein (PC00090)	35	33	38
calcium-binding protein (PC00060)	27	24	44
transfer/carrier protein (PC00219)	22	21	33
oxidoreductase (PC00176)	18	15	60
chaperone (PC00072)	24	18	34
cell adhesion molecule (PC00069)	14	10	17
transporter (PC00227)	9	8	32
transferase (PC00220)	9	7	34
extracellular matrix protein (PC00102)	7	8	14
receptor (PC00197)	8	7	18
lyase (PC00144)	4	4	13
isomerase (PC00135)	8	6	11
transcription factor (PC00218)	6	5	13
structural protein (PC00211)	3	3	7
ligase (PC00142)	2	3	9
cell junction protein (PC00070)	4	4	6
viral protein (PC00237)	1	1	1
membrane traffic protein (PC00150)	0	0	10

^a^ Note that in the PANTHER Classification System [22], proteins are classified into families and subfamilies, and families can overlap in terms of their training sequences, thus proteins from larger superfamilies could be represented in more than one PANTHER family [23].

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
