# Peer review of "In Vitro Salivary Protein Adsorption Profile on Titanium and Ceramic Surfaces and the Corresponding Putative Immunological Implications"

_ijms, 2020, doi:10.3390/ijms21093083_

Round 1
Reviewer 1 Report
The article “In vitro salivary protein adsorption profile on titanium and ceramic surfaces and the corresponding putative immunological implications” is a very interesting application of bottom-up proteomics to analyze protein adsorbed salivary proteins on conventional implant abutment materials.
The article is well written, but it shows some weak points, I will list them in this review:
- For sure the field is very interesting, but studying a system like this in a static way will give some bias: can authors discuss this problem? Will this study be similar in a more realistic situation?
- The interaction is for sure not biologically driven since ideally this material should be biologically inert. Maybe would be interesting to check the physico chemical proteins of the proteins adsorbed.
- Authors used MaxQuant but don’t talk about protein abundance. The only amount written is the total protein adsorbed but it would be interesting to see if there is protein more abundant than others.
- Authors used a pool of saliva but they did not show data for that: how different is the distribution of raw saliva vs the proteins adsorbed? In this case, would be interesting to see the quantitative values to show if there a-specifical adsorption.
Author Response
Manuscript ID: ijms-767736R Wei, C. et al. In vitro salivary protein adsorption profile on titanium and ceramic surfaces and the corresponding putative immunological implications.
Reviewer 1
The article is well written, but it shows some weak points, I will list them in this review:
1. “For sure the field is very interesting, but studying a system like this in a static way will give some bias: can authors discuss this problem?”
Authors’ reply:
The authors appreciate very much the critical comment. The static nature of the current investigation was highlighted/discussed (page 11, lines 351-352). The importance/relevance of dynamic adsorbed protein-biomaterial surface interaction is also highlighted (page 8, lines 225-232; page 11, lines 370-372; page 13, lines 483-484)
2. “…. Will this study be similar in a more realistic situation?”
Authors’ reply:
Thanks for the comment. The ‘limitations’ section is revised to reflect the experiments reported were not ‘biological’ (page 11, lines 351-355)
3. “The interaction is for sure not biologically driven since ideally this material should be biologically inert. Maybe would be interesting to check the physicochemical proteins (properties) of the proteins adsorbed.”
Authors’ reply:
Appreciate the comment. Please see reply for query #2. The need to study physicochemical properties of absorbed proteins were highlighted (page 8, lines 225-232; page 13, lines 477-480, 483-484)
4. “Authors used MaxQuant but don’t talk about protein abundance. The only amount written is the total protein adsorbed but it would be interesting to see if there is protein more abundant than others. “
Authors’ reply:
Unfortunately, during the data acquisition the quantification function was not available to us (page 8, lines 207-209; page 13, lines 455-456). The importance of quantification of the detected proteins was highlighted (page 11, line 370-372).
5. “Authors used a pool of saliva but they did not show data for that: how different is the distribution of raw saliva vs the proteins adsorbed? In this case, would be interesting to see the quantitative values to show if there a-specifical (specific) adsorption.”
Authors’ reply: Appreciate very much the comment. Protein categories from pooled filtered unstimulated saliva were added in Table 2. The similarities and differences between the saliva and adsorbed protein were described and discussed (page 4, lines 129-135; page 7, lines 182-185; page 8, lines 210-216).
Unfortunately, during the data acquisition the quantification function was not available to us (page 8, lines 207-209; page 13, lines 455-456). The importance of quantification of the detected proteins was highlighted (page 11, lines 370-372).
6. “Is the research design appropriate?”
Authors’ reply:
The Materials and Method Section was revised accordingly (page 12, lines 414-415; page 13,lines 455-456)
7. “Are the conclusions supported by the results?”
Authors’ reply:
The conclusion was revised (page 1, lines 25-27; page 13, lines 475-484)
8. “English language and style are fine/minor spell check required”
Authors’ reply:
Revised by coauthors who are native English speakers.
Reviewer 2 Report
The paper of Wei et al deals with the identification of proteins from saliva samples that attach to two different materials, titanium and ceramic. The work is well planned and the experimental part carried out schematically. The determination of proteins adsorbed on both materials give the cue to immune processes that can prompt in dental implant. The reported data can help to a better understanding on different biological phenomena involved in the first stages of adsorption as well as to improve anti-bacterial strategy and to enhance implant life. It would be interesting to study in more detail the protein-surface interaction so to rationalize the forces driving adsorption, but this can be beyond the main aim of the paper. On the whole the paper deserves publication in International Journal of Molecular Sciences, in the present form.
Author Response
Manuscript ID: ijms-767736R, Wei, C. et al. In vitro salivary protein adsorption profile on titanium and ceramic surfaces and the corresponding putative immunological implications.
Reviewer 2
1. “It would be interesting to study in more detail the protein-surface interaction so to rationalize the forces driving adsorption, but this can be beyond the main aim of the paper.”
Authors’ reply:
Appreciate the highly relevant comment. The relevance of protein-surface interaction study was highlighted/discussed (page 7, lines 192-196).
2. “For sure the field is very interesting, but studying a system like this in a static way will give some bias: can authors discuss this problem?”
Authors’ reply:
Please see reply to comments #1 and #3 to Reviewer 1.
3. “Are the methods adequately described?“
Authors’ reply:
The Materials and Method Section was revised accordingly (page 12, lines 414-415; page 13, lines 455-456)
4. “Are the conclusions supported by the results?”
Authors’ reply:
The conclusion was revised (page 1, lines 25-27; page 13, lines 475-484)
5. “English language and style are fine/minor spell check required”
Authors’ reply:
Revised by coauthors who are native English speakers.